# EEG-to-EEG: Scalp-to-Intracranial EEG Translation Using a Combination of Variational Autoencoder and Generative Adversarial Networks

**DOI:** 10.3390/s25020494

**Published:** 2025-01-16

**Authors:** Bahman Abdi-Sargezeh, Sepehr Shirani, Antonio Valentin, Gonzalo Alarcon, Saeid Sanei

**Affiliations:** 1Nuffield Department of Clinical Neurosciences, University of Oxford, Oxford OX1 2JD, UK; 2Department of Clinical Neuroscience, King’s College London, London WC2R 2LS, UK; sepehr.shirani@kcl.ac.uk (S.S.); antonio.valentin@kcl.ac.uk (A.V.); 3School of Medical Sciences, University of Manchester, Manchester M13 9PL, UK; gonzalo.alarcon@mft.nhs.uk; 4Department of Electrical and Electronic Engineering, Imperial College London, London SW7 2AZ, UK; s.sanei@imperial.ac.uk

**Keywords:** generative adversarial networks, IED detection, interictal epileptiform discharge, scalp-to-intracranial EEG translation, variational autoencoder

## Abstract

A generative adversarial network (GAN) makes it possible to map a data sample from one domain to another one. It has extensively been employed in image-to-image and text-to image translation. We propose an EEG-to-EEG translation model to map the scalp-mounted EEG (scEEG) sensor signals to intracranial EEG (iEEG) sensor signals recorded by foramen ovale sensors inserted into the brain. The model is based on a GAN structure in which a conditional GAN (cGAN) is combined with a variational autoencoder (VAE), named as VAE-cGAN. scEEG sensors are plagued by noise and suffer from low resolution. On the other hand, iEEG sensor recordings enjoy high resolution. Here, we consider the task of mapping the scEEG sensor information to iEEG sensors to enhance the scEEG resolution. In this study, our EEG data contain epileptic interictal epileptiform discharges (IEDs). The identification of IEDs is crucial in clinical practice. Here, the proposed VAE-cGAN is firstly employed to map the scEEG to iEEG. Then, the IEDs are detected from the resulting iEEG. Our model achieves a classification accuracy of 76%, an increase of, respectively, 11%, 8%, and 3% over the previously proposed least-square regression, asymmetric autoencoder, and asymmetric–symmetric autoencoder mapping models.

## 1. Introduction

Electroencephalography (EEG) is a technique used to monitor the electrical activity of the brain. This activity can be measured either on the surface of the head, known as scalp EEG (scEEG), or directly from the cerebral cortex or deeper brain regions, referred to as intracranial EEG (iEEG) [1]. scEEG is a non-invasive recording method used in clinical applications and brain computer interface [2,3,4]. scEEG signals are attenuated by the skull, resulting in a low signal-to-noise ratio, particularly when detecting low-amplitude spike-like signals such as epileptiform activity generated in deep brain areas [5,6,7]. On the other hand, iEEG, as an invasive approach, offers highly localized and detailed information due to minimal background interference and reduced signal attenuation. Therefore, it is used in clinical practice [8] and rehabilitation applications [9] whenever having high performance is crucial. Estimating the iEEG from scEEG enables the retrieval of more diagnostic information from the scEEG or clinical practices like epilepsy assessment trials.

Commonly, epilepsy is associated with seizures that strike likely due to an expected imbalance between excitation and inhibition in epileptogenic (hyperexcitable) regions in the brain [10]. The interictal recordings (between seizure onsets) involve particular neural activities called interictal epileptiform discharges (IEDs). These IEDs are identifiable through EEG as transient activities characterized by sharp waves, spikes, or poly-spikes, often succeeded by an inhibitory damped oscillation [11,12]. However, the scEEG cannot capture IED signatures, and many are not observable over the scalp [13,14,15]. Only 9% to 22% of IEDs can be visualized in scEEG recordings of patients with mesial temporal lobe epilepsy, as reported by Nayak et al. [14] and Yamazaki et al. [15].

On the other hand, iEEG is much more capable in terms of capturing epileptiform activities like IEDs without much noise contamination. The iEEG of epileptic patients suffering from temporal lobe epilepsy is recorded using foramen ovale (FO) sensors [16]. FO sensors are four- to six-contact wire sensors inserted through FO holes in the skull and placed directly on the exposed mesial temporal structures [17]. FO sensors allow for recording the iEEG and scEEG simultaneously without disrupting brain coverings [18].

By mapping the scEEG to iEEG, we can enhance the quality of scEEG in identifying IEDs [19]. We have already proposed a method using the tensor decomposition technique to project the time-frequency features of scEEG to those of iEEG in an IED detection study [20]. Multi-way analysis has extensively been used in IED detection [21,22,23,24,25]. In [20], time, frequency, sensor, and trial modes of iEEG recordings are concatenated in a four-way tensor. After decomposing the tensor into temporal, spectral, spatial, and segmental components, time-frequency features of the scEEG are projected onto the temporal components to boost the IED detection performance from the scEEG.

Several researchers have explored techniques for reconstructing iEEG from scEEG and vice versa. For instance, in a study on event-related potential (ERP) detection, Kaur et al. utilized a linear combination of intracranial ERPs to model each scalp ERP independently by employing ordinary least-squares regression [26]. Additionally, Spyrou et al. introduced a coupled scEEG-and-iEEG dictionary learning approach with sparse approximation to reconstruct iEEG from scEEG sensor data [27].

Utilizing deep neural networks to map scEEG to iEEG can be achieved by feeding scEEG data into the network to obtain the corresponding iEEG signals. Antoniades et al. developed a model based on a novel autoencoder (AE) structure to enhance the performance of scEEG-based IED detection by mapping it to iEEG [28].

Generative adversarial networks (GANs) [29] and variational autoencoders (VAEs) [30] serve as effective tools for robust data generation across various domains such as images, text, and speech. Specifically, conditional GANs (cGANs) have been developed to generate new data spaces from observed data [31,32,33,34]. In a recent work [35], we developed a cGAN-based model to map scEEG to iEEG, showing promising preliminary results.

The combination of VAEs and GANs has also emerged as a powerful approach in various medical applications due to its ability to generate realistic and diverse samples from complex high-dimensional data [36,37]. However, it has not been applied to EEG signals. In this study, we aim to design an EEG-to-EEG model termed VAE-cGAN, which combines the strengths of cGAN and VAE, to map scEEG to iEEG. In this approach, scEEG data are fed into the VAE to encode it into a latent space. Subsequently, both scEEG and the latent space are applied to the generator to produce an estimation of the iEEG signals.

## 2. EEG-to-EEG Mapping Method

Let $X\in R^{L\times M}$ and $Y\in R^{L\times \bar{M}}$ be, respectively, the scEEG and iEEG, where *L* is the number of time samples, and *M* and $\bar{M}$ are, respectively, the number of scEEG and iEEG sensors. Each observed scEEG signal is composed of a number of iEEG sources plus noise. Their relationship can be modeled as(1)X=F(Y)+noise,
where F is a non-linear mapping function between scEEG and iEEG.

In our proposed method, namely scalp-to-intracranial EEG translation, a combination of VAE and cGAN, named as VAE-cGAN, is used as the mapping model. The overview of our networks is shown in Figure 1. The following sub-sections firstly give a general overview of VAE and GAN. Then, the details of the proposed scalp-to-intracranial EEG translation are provided.

### 2.1. VAE

A VAE is made up of two networks, namely encoder and decoder, where the encoder network E encodes a data sample X into a latent representation z estimated from the distribution, and the decoder network (here, it performs like generator G in GAN) decodes the latent representation to reconstruct the data (called generated data $\tilde{Y}$) with minimum error.

A VAE imposes a prior p(z) over the latent space to ensure that it follows a Gaussian distribution, ϵ∼N(0,I). In other words, rather than mapping the data sample X directly to z, the encoder network E maps X into two different vectors that are the mean μ and the standard deviation (STD) σ of multivariate Gaussian distribution. Then, the Gaussian sample ϵ is scaled by μ and σ as follows:(2)z=μ+ϵ⊙σ,
where ϵ∼N(0,I) is an auxiliary noise variable, and ⊙ indicates the element-wise product.

To enforce the encoder to map the data sample X into a Gaussian distribution, the Kullback–Leibler (KL) divergence $D_{KL}$ is calculated as follows:(3)$$L_{D_{KL}}=D_{KL}\left(E\left(X\right)∥p\left(z\right)\right).$$

To estimate the objective loss in (Equation 3), the reparameterization trick [30] is applied, yielding the following expression:(4)$$L_{D_{KL}}≃-\frac{1}{2}\sum_{z=1}^{Z}\left(1+log\left(\sigma_{z}^{2}\right)-\mu_{z}^{2}-\sigma_{z}^{2}\right),$$
where *Z* is the dimension of the latent space.

### 2.2. GAN

A GAN consists of a generator G and a discriminator D. The generator G maps the latent space z with a prior probability distribution p(z) to a data space to generate a data sample. The discriminator D takes either a real or a generated data sample as input and predicts a binary class of real or fake (generated). An adversarial loss is employed to train the generator and discriminator. In most studies [38,39,40] including the original GAN [29], the binary cross entropy is used in a min-max game approach as loss function:(5)$$\begin{array}{ccc} \; & L_{GAN}= & E_{Y}\left[log\left(D\left(Y\right)\right)\right]+\\ \; & \; & E_{z}\left[log\left(1-D\left(G\left(X,z\right)\right)\right)\right], \end{array}$$
where G minimizes the objective loss function against an adversarial D maximizing it. However, instead of binary cross entropy, the hinge loss is employed here as an adversarial loss, which is explained latter.

### 2.3. Scalp-to-Intracranial EEG Translation

Inspired by image-to-image translation methods [39,40,41,42,43] and Speech Enhancement GAN [34], which are based on cGANs, the proposed scalp-to-intracranial EEG translation method maps the scEEG to the iEEG using the proposed cGAN. We combine the generator G of our cGAN with VAE, called VAE-cGAN, by letting them share their parameters and train jointly. Apart from differences in structures, loss functions, and their applications [44,45,46], the main difference between the previously developed GAN-VAE with our proposed VAE-cGAN is that, in our proposed method, both input data sample X and the encoded latent space z are given to the generator G, while in others the input of the generator is only the latent space z.

In our scalp-to-intracranial EEG translation VAE-cGAN model, the encoder E encodes the scEEG X sensor data to the latent space z. The objective of the generator G is to translate/map the scEEG X as well as the latent space z to the iEEG Y, where the translated signal is named estimated iEEG $\tilde{Y}$. On the other hand, the goal of the discriminator D is to distinguish the iEEG Y from the estimated iEEG $\tilde{Y}$. As a result, VAE-cGAN is(6)$$\begin{array}{c} z=E(X)\;\;\;\;\;\;\;\; \end{array}$$(7)$$\begin{array}{c} \tilde{Y}=G\left(X,z\right), \end{array}$$
where $\tilde{Y}\in R^{L\times \bar{M}}$ is the estimated iEEG.

#### 2.3.1. Encoder

The scEEG X is given as the input to the encoder network. As shown in Figure 2, our encoder E consists of sequences of convolutional layers with the filter size of $\left(l_{e}\times 1\right)$ and stride of (2×1), followed by the instance normalization [47] and leaky ReLU layers. The output of the last convolutional layer is projected onto a couple of dense layers, which are known as the mean μ and STD σ of Gaussian distribution for sampling the latent code. Finally, the latent space which has a dimension of z is acquired by scaling and shifting the Gaussian distribution ϵ∼N(0,I) using μ and σ, as demonstrated in (Equation 2).

#### 2.3.2. Generator

The generator G is developed using the SPatially-Adaptive (DE)normalization (SPADE) layer [41], which serves as a cornerstone for effective conditional data normalization. Unlike traditional normalization methods that depend solely on the internal statistics of activations, SPADE introduces an external conditioning mechanism, enabling the normalization process to adapt dynamically based on external inputs. In this study, the external data, scEEG X, provide the modulation parameters γ and β, which are used to denormalize the activations after they are normalized to have zero mean and unit standard deviation (STD) [41,48,49].

The SPADE block is shown is Figure 3A. In SPADE, the activation is normalized in the sensor wise manner, like batch normalization [50]. Then, it is multiplied by γ and added to β element-wise:(8)$$SPADE_{out}=\gamma \times A+\beta ,$$
where A is the normalized activation layer, and γ and β are the learned modulation parameters of the normalization layer whose parameters are inferred from scEEG X.

SPADE’s primary advantage lies in its ability to preserve spatial information. In conventional normalization layers such as instance normalization [47], spatial information is often lost because these methods normalize activations uniformly across spatial dimensions, disregarding location-specific features. In contrast, SPADE preserves spatial information by normalizing only the activation layer from the previous step, while the scEEG is directly used to modulate the activations through spatially adaptive parameters (γ or β). This is critical for mapping scEEG to iEEG, as the spatial relationships among sensors carry essential information about the underlying brain activity. By conditioning on the scEEG data, SPADE ensures that spatial dependencies are retained, which is particularly important for accurate modeling of neural signal distributions. Furthermore, by employing the learned modulation parameters based on external input, SPADE enables the generator to adapt dynamically to variations in scEEG signals. This adaptability enhances the model’s robustness, making it suitable for diverse recording conditions and patient-specific data variations.

SPADE is implemented within a ResNet-based architecture, forming SPADE ResNet blocks (Figure 3B), comprising two SPADE blocks which are followed by hyperbolic tangent (Tanh) activation and convolutional layers. These blocks combine the benefits of SPADE with the residual learning framework, allowing the generator to effectively capture both spatial and temporal complexities of the input signals. When the input and output dimensions differ, SPADE replaces the standard skip connection to ensure dimensional consistency.

The generator architecture is demonstrated in Figure 4. The latent space z is first passed through a dense (fully connected) layer to match its dimensions with those required for the first spatially adaptive modulation (γ or β). After normalization, together with downsampled scEEG, it is fed to the SPADE ResNets. By integrating the latent space with scEEG inputs in the generator, the model combines a globally informed representation (from z) with localized spatial features (from X).

SPADE ResNets are followed by upsampling layers with the size of (2,1). After the series of SPADE ResNets and upsampling layers, the generator incorporates long short-term memory (LSTM) layers to model the temporal dynamics of EEG signals. The inclusion of LSTM layers is pivotal for ensuring that the generator can handle the inherently time-varying nature of EEG data. LSTM layers excel at modeling long-term dependencies in sequential data, making them particularly well-suited for analyzing EEG signals. By processing the data over *L* time steps, the LSTM layers capture intricate temporal relationships between scEEG and iEEG signals, ensuring that the generated iEEG signals reflect realistic and clinically relevant temporal dynamics. The second LSTM layer is followed by a time-distributed dense layer, which generates outputs for each time step individually. This layer ensures that the final output $\tilde{Y}$ maintains the same temporal and spatial dimensions as the target iEEG Y. The tanh activation function applied in this stage further aids in producing smooth and bounded outputs, critical for accurately representing neural activity.

#### 2.3.3. Discriminator

In our study, a Markovian discriminator is employed [51]. For these discriminators, the dimension of output is not 1, it is rather $p_{1}\times p_{2}$, called patch, and the discriminator determines whether each patch is real or not [51,52].

The input to the discriminator comprises either the actual iEEG data Y or the estimated iEEG data $\tilde{Y}$, both having *L* time samples and $\bar{M}$ source observations. Here, the discriminator pipeline consists of a sequence of convolutional layers. With the exception of the final convolutional layer, all convolutional layers are configured with a filter size of $\left(l_{d}\times 1\right)$ and a stride of (2×1). These convolutional layers are subsequently followed by IN and leaky ReLU layers. The last convolutional layer does not have normalization and activation layers. The patch size of the discriminator is set to $1\times \bar{M}$. This means that the discriminator tries to classify each sensor as either the real iEEG or the estimated iEEG.

#### 2.3.4. Optimization and Loss Function

Considering the dataset and application, a variety of loss functions have been used to train the discriminator and generator in GANs. In this context, we employ the hinge loss, a loss function previously utilized in adversarial approaches [41,53,54], to train the discriminator D:(9)$$\begin{array}{c} \begin{array}{ccc} & L_{D}= & -E_{Y}\left[min\left(0,-1+D\left(Y\right)\right)\right]\\ & & -E_{X,z}\left[min\left(0,-1-D\left(G\left(X,z\right)\right)\right)\right] \end{array} \end{array}$$(10)$$\begin{array}{c} \begin{array}{ccc} & L_{G_{h}}= & -E_{X,z}\left[D\left(G\left(X,z\right)\right)\right]. \end{array} \end{array}$$

In addition to the hinge loss, other three loss functions—namely KL divergence loss $L_{D_{KL}}$, $L_{1}$ loss $L_{L_{1}}$, and feature matching loss $L_{FM}$—are coupled together and used to optimize the generator:(11)$$L_{G}=L_{G_{h}}+\lambda_{1}L_{D_{KL}}+\lambda_{2}L_{L_{1}}+\lambda_{3}L_{FM},$$
where $\lambda_{1}$, $\lambda_{2}$, and $\lambda_{3}$ are penalty terms.

$L_{D_{KL}}$ is calculated using (Equation 4). $L_{1}$ loss is defined as the distance between the real data sample Y and the generated data $\tilde{Y}$, estimated as follows:(12)$$L_{L_{1}}=E_{\left(\tilde{Y},Y\right)}[∥Y-\tilde{Y}{∥}_{1}].$$

Feature matching loss $L_{FM}$ has been used in image-to-image studies to improve the adversarial loss [40,55]. The feature matching loss, denoted as $L_{FM}$, is derived from the discriminator and facilitates the learning of global information by the GAN module through multi-scale features. This loss is obtained by comparing the intermediate feature maps of the actual intracranial EEG Y with those of the estimated intracranial EEG $\tilde{Y}$. Let $D^{\left(i\right)}$ represent the *i*-th layer in the discriminator D. The feature matching loss $L_{FM}$ is computed as follows:(13)$$L_{FM}=E\sum_{i=1}^{I_{D}}\frac{1}{F_{i}}{\left|D^{\left(i\right)}\left(Y\right)-D^{\left(i\right)}\left(\tilde{Y}\right)\right|}_{1},$$
where $I_{D}$ is the total number of layers in the discriminator and $F_{i}$ is the number of features in layer *i*.

## 3. Experiment

The proposed method is used to translate the scEEG to iEEG recordings. After mapping, the IEDs are detected from the estimated iEEG.

### 3.1. Dataset

We utilize a concurrent EEG dataset in which the scEEG and iEEG sensor data were simultaneously recorded from 18 persons diagnosed with temporal lobe epilepsy at King’s College London Hospital. Among these patients, the seizure onset was localized to mesial temporal structures in 10 cases, while in the remaining 8 patients, the onset was identified in the lateral temporal region. This retrospective study is based on standard medical recordings from patients at King’s College Hospital, and it did not need approval by the ethics committee.

Twenty (20) scalp sensors and twelve (12) FO sensors (six-contact wire sensors for each lateral) were used, respectively, for recording the scEEG and iEEG with the sampling rate of 200 Hz. Both scEEG and iEEG were band-pass filtered between 0.3 and 70 Hz. More details are given in our previous study [56].

### 3.2. IED Annotation and Pre-Processing

An experienced neurologist annotated the IEDs according to the spatial distribution and morphological characteristics of the iEEG, meaning that the iEEG sensor recording was used as a ground truth in the IED annotation. Table 1 shows the number of IEDs observed in the iEEG, the number of IEDs visible over the scEEG as spikes or small transient activities, and the percentage of visible IEDs for each patient. Only 18.8% of IEDs are visible over the scalp.

IED segments used for mapping and classification were set to a fixed length of 320 ms (64 time samples), including 160 ms before and after the IED onset, irrespective of the actual IED duration. Non-IED segments were randomly selected from time periods without annotated IEDs, ensuring an equal number of IED and non-IED segments for analysis.

Both scEEG and iEEG sensor recordings were passed through a high-pass filter with a cutoff frequency of 1 Hz. In addition, a 50 Hz notch filter was employed to eliminate the power line interference. Furthermore, a spatial filter, common average reference, is applied to the scEEG to eliminate the artifacts.

### 3.3. Translating/Mapping scEEG to iEEG

The proposed scEEG-to-iEEG translation method is employed to map scEEG to iEEG. The complete architecture is shown in Figure 5. In the training stage, the scEEG with 64 time samples and 20 sensors is projected onto a latent space z with the dimension of 256. Then, the latent space, as well as the scEEG sensor data, are fed to the generator G to generate an estimation of the corresponding iEEG sensor data. Finally, the estimated iEEG and the real iEEG are given to a patch discriminator to be classified as real or unreal.

In this study, both scEEG and iEEG were directly fed into the networks without undergoing artifact rejection, as the scEEG and iEEG recordings were artifact-free (artifactual periods were excluded). However, like most deep learning-based methods, our algorithm is sensitive to noise. Significant alterations in the input data distribution, such as those introduced by artifacts, can adversely impact model performance. To address this challenge, artifact rejection techniques such as independent component analysis [57], canonical correlation analysis [58], and spatio-spectral component analysis [59] can be utilized to enhance the signal-to-noise ratio of EEG signals.

### 3.4. Classification and Cross Validation

The EEGNet [60] has been proven effective in EEG classification [60,61]. For classifying IEDs and non-IEDs, an EEGNet with minor modifications is employed. The batch normalization in EEGNet layers has been eliminated from the classification network. In addition, Max Pooling is used instead of Average Pooling.

The IEDs are detected using two different approaches: intra-subject and inter-subject. In the intra-subject classification approach, the data of a subject are divided into training (70%), validation (10%), and test datasets (20%). On the other hand, in the inter-subject classification, the data from *N* subjects are used for training, and the data of another subject for testing. Here, *N* is the number of subjects whose IEDs are detected with high accuracy (>70%) in the intra-subject classification approach. This approach is repeated for all 18 subjects.

In the inter-subject classification approach, the scEEG of all *N* training subjects and the test subject are mapped to the iEEG using each of the trained VAE-cGAN ($G_{n},\;\;n=\left\{1,2,\ldots ,N\right\}$), to obtain the estimated iEEG, $X\overset{G_{n}}{\to }{\tilde{Y}}_{n}$. Then, each of the estimated iEEG ${\tilde{Y}}_{n}$ is given to the EEGNet to classify IEDs and non-IEDs. Finally, to find the segment labels in the test data, the output probabilities of *N* EEGNets are averaged (average voting classification). Figure 6 shows the diagram of the inter-subject classification approach.

## 4. Results

Our proposed VAE-cGAN technique is compared with three previously developed methods [26,28,35]. In [26], the authors modeled the scEEG from the iEEG using a least-square regression (LSR) to detect scalp ERPs. Antoniades et al. designed two AE-based methods, namely asymmetric AE (AAE) and asymmetric–symmetric AE (ASAE), for mapping the scEEG to iEEG in an IED detention study [28]. Since the scEEG and iEEG had different dimensions and consequently the input and output size of AE were different, the AE is called AAE. In ASAE, after mapping the scEEG to iEEG using AAE, its output was mapped onto the iEEG again using a symmetric AE. This method used two asymmetric and symmetric AEs, named as ASAE. In [35], we reported some preliminary results of a cGAN-based model deployed for mapping scEEG to iEEG.

### 4.1. The Performance of Mapping Model

The scEEG, iEEG, and estimated iEEG are shown in Figure 7. In Figure 7A, the samples are averaged across all sensors of the segment, while Figure 7B presents a single sensor of a random segment. Based on the figures, the estimated iEEG closely mirrors the actual iEEG in samples containing IEDs. Although epileptic spikes are not visible in the scEEG, the estimated iEEG accurately replicates the genuine iEEG, clearly displaying the presence of IEDs. In cases without IEDs, the estimated iEEG roughly tracks the pattern of the iEEG.

To show the performance of the mapping model numerically, the mean squared error (MSE), cosine similarity (COSSIM), and Pearson correlation coefficient (PCORR) are obtained. MSE is calculated for each sensor as follows:(14)$$MSE=\frac{1}{L}\sum_{l=1}^{L}{\left(y_{l}-{\tilde{y}}_{l}\right)}^{2},$$
where y and $\tilde{y}$ are, respectively, a single sensor of iEEG and the corresponding estimated iEEG.

COSSIM demonstrates how well the real iEEG and the estimated iEEG are similar to each other. Ranging from −1 to 1, COSSIM yields 1 for identical vectors and −1 for completely opposite (in phase) vectors as follows:(15)$$COSSIM=\frac{\sum_{l=1}^{L}y_{l}{\tilde{y}}_{l}}{\sqrt{\sum_{l=1}^{L}y_{l}^{2}}.\sqrt{\sum_{l=1}^{L}{\tilde{y}}_{l}^{2}}}.$$

The PCORR is a statistical measurement that gives the linear correlation between two continuous variables. It ranges between −1 and 1, where 1 indicates a perfect positive linear correlation and −1 represents a perfect negative linear correlation:(16)$$PCORR=\frac{cov(y,\tilde{y})}{\sigma_{y}\sigma_{\tilde{y}}}$$
where cov is the covariance and $\sigma_{y}$ and $\sigma_{\tilde{y}}$ are, respectively, the standard deviation of y and $\tilde{y}$.

While the mapping performance is not reported in the studies under comparison, Table 2 exclusively presents the performance metrics of our mapping model. It demonstrates an MSE of 0.014, PCORR of 0.35, and COSSIM of 0.34.

### 4.2. The Performance of IED Detection

Accuracy (ACC), sensitivity (SEN), specificity (SPC), precision (PRC), F1-score (F1-S), receiver operating characteristic (ROC) curve, and the area under ROC curve (AUC) are obtained as the evaluation criteria for the inter-subject classification approach. The IED class is considered as positive and the non-IED class as negative. As a result, SEN shows the model ability in detecting the IEDs, while SPC indicates the model ability in detecting the non-IEDs.

The ACC of our proposed VAE-cGAN and the compared methods (LSR, AAE, ASAE, cGAN) are presented in Table 3, obtained in both the intra- and inter-subject classification approaches. In the inter-subject classification approach, our proposed VAE-cGAN outperforms other methods, achieving an ACC of 69%. This is, respectively, 7% and 3% higher than the ACC values obtained using LSR and AAE, while ASAE and cGAN provide comparable ACC values of 68%. In the intra-subject classification approach, VAE-cGAN achieves an ACC of 77%. This represents an improvement of 12% over LSR, 9% over AAE, 3% over ASAE, and 1% over cGAN.

Figure 8A displays ACC, SEN, SPC, F1-S, and AUC values in the inter-subject classification approach, averaged across subjects. In terms of ACC, SEN, and F1-S, VAE-cGAN demonstrates the best performance. In terms of SPC, cGAN provides the best value. Both VAE-cGAN and ASAE obtain the best AUC value of 0.74. The ROC curve is also shown in Figure 8B.

### 4.3. Ablation Study

To find out how the encoder and SPADE ResNet affect the model performance, ablation studies are carried out. To investigate the impact of the encoder, normalized Gaussian noise is used as the latent space variable, referred to as No-Encoder. The impact of SPADE ResNet is investigated by replacing it with a convolutional layer, named as No-SPADE-ResNet.

ACC, MSE, PCORR, and COSSIM are obtained for the intra-subject classification approach, illustrated in Table 4. For the No-Encoder approach, the ACC decreases to 70%, while the No-SPADE-ResNet approach provides the ACC of 64%. In addition, both PCORR and COSSIM decrease for both No-Encoder and No-SPADE-ResNet approaches. MSE increases to 0.15 and 0.17, respectively, for the No-Encoder and No-SPADE-ResNet approaches.

## 5. Discussion

The quest to extract richer diagnostic insights from noninvasive sensor recordings remains a primary focus for clinicians and researchers. Transitioning from invasive iEEG to noninvasive scEEG while maintaining diagnostic efficacy offers numerous advantages, considering the ease and accessibility of scEEG sensor recording procedure compared to iEEG. Here, we propose a method based on VAE and cGAN to map scEEG data to iEEG counterparts to improve the performance of the IED detection model.

Most of the studies in which the IEDs were annotated from only scEEG reported high performance [24,25,62,63]. The reason is that they only detect the scalp-visible IEDs, which include a maximum of 22% of all IEDs [14,15]. However, for detecting all IEDs, iEEG should be used as the ground truth for annotating IEDs. This has been done in a few studies [21,23,56,64]. They showed poorer performance given that, in these studies, the accuracy measure considers all the IEDs, not just scalp-visible IEDs.

In our previous work, we employed tensor factorization to map scEEG time-frequency features to iEEG, enhancing IED detection from scEEG [20]. Notably, our intra-subject approach achieved an accuracy of 72%, surpassing the performance of deep learning-based methods proposed in our current study and in [28,35]. It is important to note that the subjects and trials utilized in [20] differ from those in our current study and in [28,35]; hence, a direct comparison of results was not conducted.

Our method, along with the models introduced in [28,35], offers the significant advantage of generating iEEG estimations from scEEG data. IEDs are well-established biomarkers of epilepsy [65], as their presence strongly predicts seizure recurrence following a first seizure [66,67], thereby playing a critical role in supporting epilepsy diagnoses. In many cases, relying solely on ictal events for diagnosis and monitoring can be time-consuming due to their low frequency. This underscores the importance of IEDs in the diagnosis and classification of epileptic cases. Furthermore, accurate IED detection serves as a valuable guideline for pre-ictal state monitoring, treatment planning, and surgical decision-making. Therefore, detecting 69% of IEDs using VAE-cGAN represents a significant improvement over the 18.8% detection rate achieved through scEEG visualization (see Table 1), making it more useful for clinical applications.

Although the proposed VAE-cGAN has higher computational complexity than the compared models, it remains feasible for practical applications. The training of the proposed VAE-cGAN model was performed on a macOS system with an M2 Max series chip and 32 GB of memory. The training time varies with the dataset size, taking approximately 15 min for a single subject with a moderate number of trials. While this computational complexity is acceptable, it can be further optimized using high-performance machines. Once the model is trained, it requires only a few milliseconds to generate iEEG from scEEG, demonstrating its efficiency in inference. Furthermore, these models are primarily intended for offline data analysis rather than real-time applications. In such scenarios, main priorities are the accuracy and robustness of the algorithm rather than the computation cost, making VAE-cGAN a viable and effective choice.

The efficient application of mapping models for comprehensive iEEG analysis relies heavily on the consistent data acquisition from specific neuroanatomical regions across patient cohorts, such as the mesial temporal lobe in epilepsy or the subthalamic nucleus (STN) in Parkinson’s disease. This focus on neuroanatomical specificity highlights their crucial role in uncovering disorder-specific pathophysiological mechanisms and informing tailored therapeutic interventions, thereby advancing both our understanding and treatment of neurological conditions. However, the reliance on precise neuroanatomical data poses a limitation when attempting to generalize a mapping model across diverse clinical applications. Variations in patient-specific anatomy, electrode placement, and the heterogeneity of neurological disorders may reduce the model’s adaptability, emphasizing the need for robust and flexible approaches that can accommodate these differences without compromising accuracy or clinical utility.

Although VAE-cGAN is not yet generalized for direct use in other EEG-to-EEG applications, it holds significant potential for adaptation to various clinical practices. For instance, in adaptive deep brain stimulation systems for Parkinson’s disease, the recordings from STN are utilized to detect pathological activity that guides stimulation delivery [68,69]. However, these recordings are often contaminated with artifacts due to the close proximity of the recording and stimulation sites. By extending VAE-cGAN to leverage scEEG data for detecting pathological STN activity, it would be possible to overcome this limitation.

There remains significant potential to enhance the performance of mapping models. One promising strategy for improving the accuracy of VAE-cGAN in estimating iEEG and detecting IEDs is through the acquisition of prolonged concurrent iEEG and scEEG recordings. Extended recordings capture a more comprehensive and detailed representation of neuronal activity, enabling deep learning models to better learn the underlying patterns. Furthermore, exploring innovative regularization techniques and transfer learning may mitigate the data demands, enabling their deployment even in scenarios with limited training data. By addressing these challenges, mapping models can achieve greater precision and robustness, paving the way for more effective clinical applications.

## 6. Conclusions

In this study, we introduce VAE-cGAN as a method to map scEEG to iEEG sensor recordings obtained from epileptics with epileptic spikes/IEDs, comparing its performance against established techniques [26,28,35]. VAE-cGAN demonstrates superior ACC, SEN, and SPC when compared to LSR, AAE, ASAE, and cGAN. It achieves an average ACC of 69% for the inter-subject classification approach which is, respectively, 7%, 3%, 1%, and 1% higher than ACC values obtained using LSR, AAE, ASAE, and cGAN. In addition, the VAE-cGAN outperforms the compared methods in the intra-subject classification approach by achieving the average ACC of 77% which is, respectively, 12%, 9%, 4%, and 1% higher than ACC values achieved using LSR, AAE, ASAE, and cGAN. However, here the trade-off is between the higher accuracy of the system and its high computational complexity as for other deep neural networks.

## Figures and Tables

**Figure 1 sensors-25-00494-f001:**
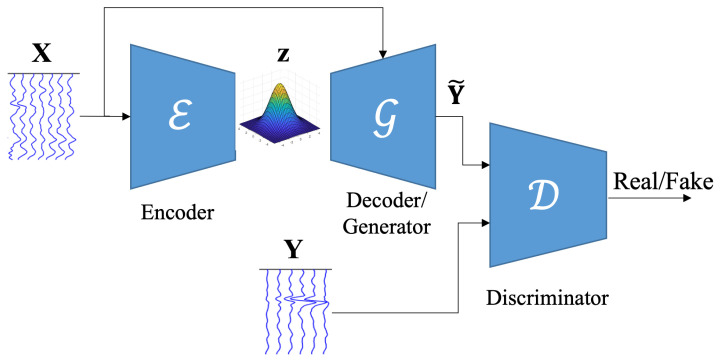
The overviewof our proposed EEG-to-EEG network. X and Y are, respectively, the scEEG and iEEG. scEEG is fed to the encoder to be encoded to a latent space, which, along with scEEG, serves as an input to the generator. The generated data and the real iEEG are provided to the discriminator to be classified as real or fake.

**Figure 2 sensors-25-00494-f002:**
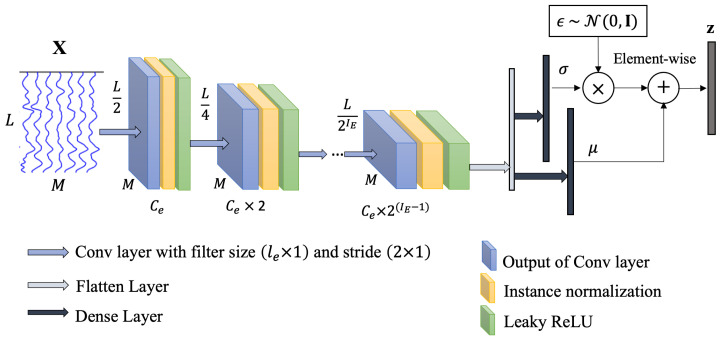
The encoder network E. The input data, X, are scEEG. $C_{e}$ represents the coefficient of the number of filters for the Conv layer, and $I_{E}$ represents the total number of layers for the encoder. μ and σ indicate, respectively, the mean and standard deviation of multivariate Gaussian distribution. z is the latent space.

**Figure 3 sensors-25-00494-f003:**
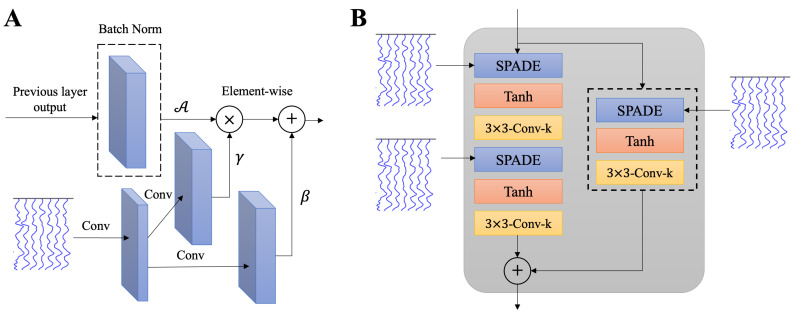
(**A**) The SPADE block. In the SPADE block, the scEEG is first projected onto an embedding space and then convolved to produce the modulation parameters γ and β. After normalizing the activation layer, A, it is multiplied by γ and added to β element-wise. The SPADE is firstly proposed in [41]. (**B**) The SPADE ResNet consisting of two SAPDE blocks is followed by a Tanh activation and convolutional layers.

**Figure 4 sensors-25-00494-f004:**
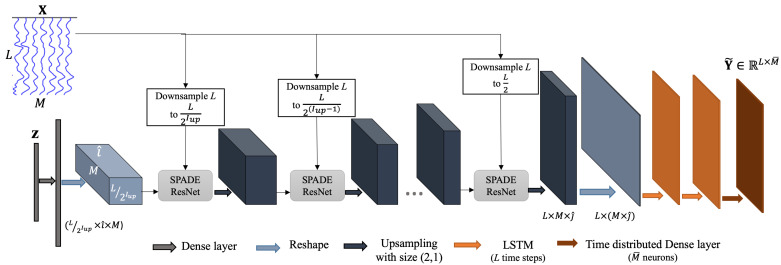
The generator network G. scEGG, X, and the latent space, z, serve as input to the generator. *L* and *M* are, respectively, the number of time samples and channels of scEEG. $I_{up}$ is the number of upsampling or SPADE ResNet layers. The output, $\tilde{Y}$, is the synthetic iEEG.

**Figure 5 sensors-25-00494-f005:**
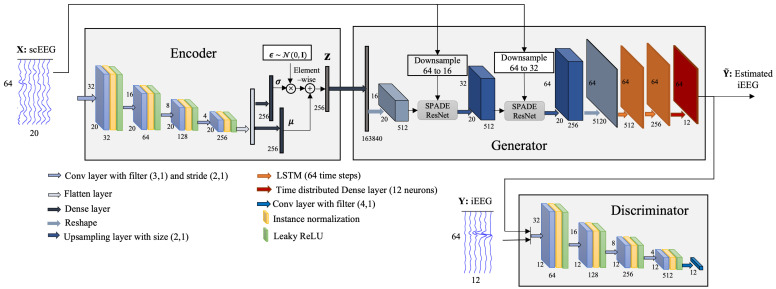
The proposed VAE-cGAN model for mapping the scEEG to iEEG. scEEG is encoded to a latent space. Then, the latent space alongside scEEG is fed to the generator to generate synthetic iEEG. The estimated iEEG and real iEEG are given to the discriminator to be classified as real or fake.

**Figure 6 sensors-25-00494-f006:**
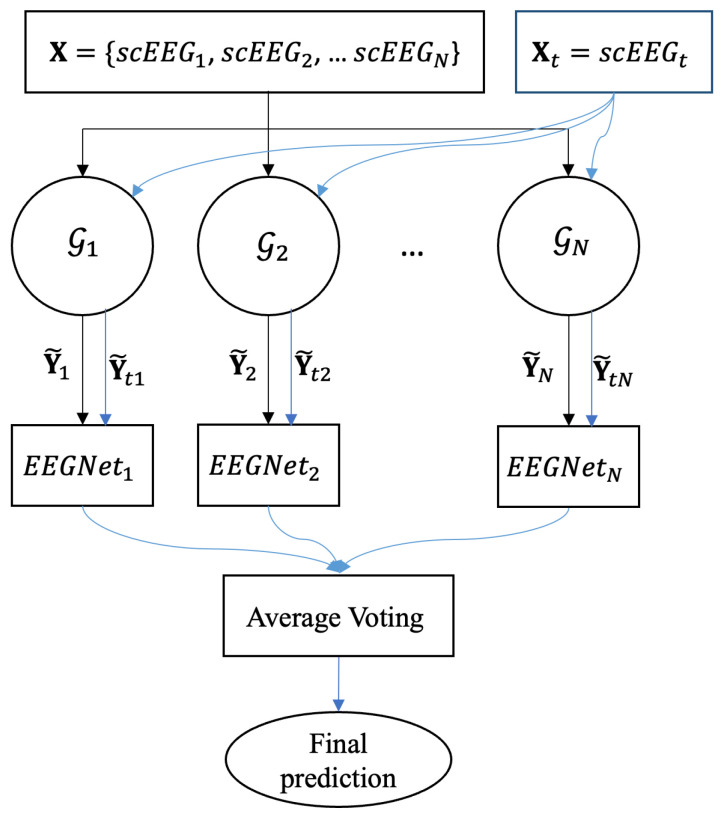
The diagram of the inter-subject classification approach.

**Figure 7 sensors-25-00494-f007:**
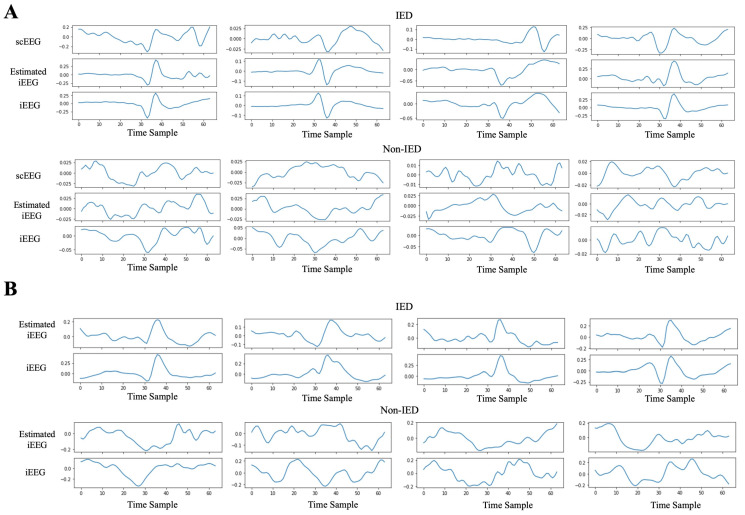
(**A**) Samples of IEDs (top) and non-IEDs (bottom) averaged across all sensors are depicted. The scEEG, iEEG, and stimated iEEG generated by VAE-cGAN are illustrated. (**B**) Samples of IEDs (top) and non-IEDs (bottom) for a single sensor. The scEEG and estimated iEEG using VAE-cGAN are shown. In both (**A**,**B**), estimated iEEG follows the trend of iEEG in the IED samples. The IEDs start in point sample 32, and the sampling rate is 200 samples/s.

**Figure 8 sensors-25-00494-f008:**
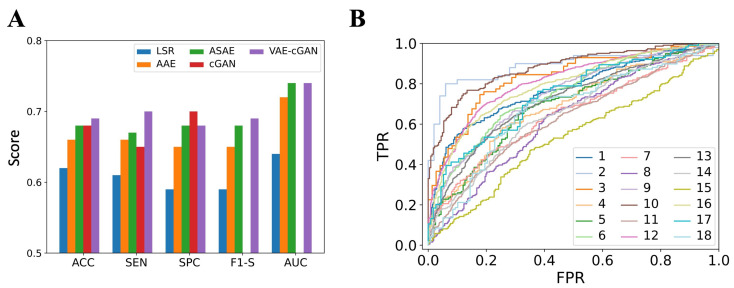
(**A**) Comparison of ACC, SEN, SPC, F1-S, and AUC metrics between our proposed VAE-cGAN method and the benchmarked LSR, AAE, ASAE, and cGAN methods in the inter-subject classification scenario. Notably, F1-S and AUC values were not available for the cGAN method. (**B**) The ROC curve provided by VAE-cGAN.

**Table 1 sensors-25-00494-t001:** Total number of IEDs and scalp-visible IEDs recorded for each subject, with the percentage of scalp-visible IEDs relative to the total IEDs shown in parentheses.

Sub.	No. of IEDs	No. of Scalp-Visible IEDs (Their Percentage from All IEDs)
S1	342	129 (37.7%)
S2	50	17 (34.0%)
S3	71	9 (12.7%)
S4	165	60 (36.4%)
S5	158	19 (12.0%)
S6	472	77 (16.3%)
S7	199	45 (22.6%)
S8	317	143 (45.1%)
S9	341	46 (13.5%)
S10	224	86 (38.4%)
S11	848	90 (10.6%)
S12	953	12 (1.3%)
S13	829	35 (4.2%)
S14	536	74 (13.8%)
S15	260	28 (10.8%)
S16	606	11 (1.8%)
S17	114	28 (24.6%)
S18	118	5 (4.2%)
Mean	366.8	50.8 (18.8%)

**Table 2 sensors-25-00494-t002:** The mapping performance of our proposed VAE-cGAN method for the intra-subject classification approaches. MSE, PCORR, and COSSIM are presented.

Subject	MSE	PCORR	COSSIM
S1	0.012	0.48	0.47
S2	0.012	0.46	0.45
S3	0.017	0.21	0.21
S4	0.013	0.32	0.32
S5	0.014	0.33	0.33
S6	0.014	0.31	0.31
S7	0.017	0.22	0.21
S8	0.022	0.18	0.18
S9	0.013	0.38	0.37
S10	0.013	0.32	0.32
S11	0.015	0.33	0.32
S12	0.012	0.53	0.53
S13	0.014	0.36	0.35
S14	0.013	0.42	0.41
S15	0.015	0.23	0.23
S16	0.012	0.46	0.45
S17	0.013	0.28	0.28
S18	0.015	0.22	0.21
Mean	0.014	0.35	0.34

**Table 3 sensors-25-00494-t003:** Comparison of IED detection ACC between our proposed VAE-cGAN method and the benchmarked LSR, AAE, ASAE, and cGAN methods across both intra- and inter-subject classification approaches. Values in parentheses represent intra-subject classification results. All values are presented in percentages (%).

Subject	LSR [26]	AAE [28]	ASAE [28]	cGAN [35]	VAE-cGAN
S1	65 (72)	85 (80)	87 (78)	67 (78)	71 (80)
S2	86 (81)	92 (82)	94 (88)	83 (95)	87 (95)
S3	65 (69)	72 (72)	69 (82)	74 (90)	77 (78)
S4	58 (62)	58 (71)	59 (77)	66 (81)	64 (74)
S5	55 (55)	64 (64)	65 (75)	67 (73	67 (74)
S6	61 (59)	70 (60)	71 (63)	68 (68)	70 (74)
S7	59 (64)	54 (62)	67 (72)	64 (67)	62 (68)
S8	55 (66)	55 (62)	57 (68)	63 (72)	61 (72)
S9	63 (65)	61 (74)	62 (68)	61 (71)	68 (77)
S10	66 (70)	71 (65)	74 (77)	75 (91)	82 (90)
S11	63 (64)	65 (67)	65 (68)	61 (62)	60 (63)
S12	73 (79)	75 (84)	77 (84)	79 (84)	75 (84)
S13	62 (71)	62 (72)	64 (71)	63 (74)	67 (75)
S14	59 (62)	66 (71)	67 (65)	63 (69)	60 (72)
S15	50 (46)	50 (53)	50 (52)	55 (59)	55 (60)
S16	51 (55)	67 (77)	68 (72)	75 (77)	73 (86)
S17	54 (62)	59 (54)	62 (71)	66 (78)	69 (86)
S18	66 (64)	61 (53)	67 (75)	65 (72)	67 (67)
Mean	62 (65)	66 (68)	68 (73)	68 (76)	**69 (77)**

Our method outperfoms the compared methods in IED detection.

**Table 4 sensors-25-00494-t004:** The performance of the ablation studies where the impact of encoder and SPADE ResNet are investigated. ACC, MSE, PCORR, and COSSIM are obtained for the intra-subject classification approach.

Ablation	ACC	MSE	PCORR	COSSIM
No-Encoder	70	0.15	0.32	0.32
No-SPADE-ResNet	64	0.17	0.26	0.25

## Data Availability

Anonymized datasets used for the current study and computer code are available from the corresponding authors upon request.

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
