# Peer review of "EEG-to-EEG: Scalp-to-Intracranial EEG Translation Using a Combination of Variational Autoencoder and Generative Adversarial Networks"

_sensors, 2025, doi:10.3390/s25020494_

Round 1
Reviewer 1 Report
Comments and Suggestions for Authors
Your article proposes a deep learning model called VAE-cGAN for translating scalp electroencephalography (scEEG) signals to intracranial electroencephalography (iEEG) signals.Since scEEG signals are noisy and have low resolution, while iEEG signals have high resolution, this model can enhance the quality of scEEG signals and improve the ability to identify interictal epileptiform discharges (IEDs).Experimental results show that VAE-cGAN achieves higher accuracy in IED detection compared to previously proposed methods.
The topic is interesting, but there are some major concerns that that need to be addressed:
-
Model Explanation: The article could provide a more detailed explanation of the VAE-cGAN model’s principles and structure. For instance, it could elaborate on the role of SPADE blocks and LSTM layers, and how scEEG and latent space are inputted into the generator.P4P5
-
Comparative Experiments: The article could include additional comparative experiments, such as comparisons with other deep learning models or tests on different datasets, to further validate the effectiveness and generalization ability of VAE-cGAN.
-
Computational Complexity: The article could analyze the computational complexity of VAE-cGAN and compare it with other methods to assess its feasibility in practical applications.
-
Clinical Applications: The article could further explore the potential clinical applications of VAE-cGAN, for instance, discussing how it could be used for epilepsy diagnosis and treatment.
-
Model Limitations: The article could discuss the limitations of VAE-cGAN, such as its potential sensitivity to noise or difficulty in handling individual differences between patients.
-
Future Work: The article could propose some future research directions, for example, exploring ways to improve the performance of VAE-cGAN or applying it to other EEG-related tasks.
-
It is recommended to cite the following articles to improve the content of the paper and make it more comprehensive:
â‘ W. Cheng, S. Zhang and Y. Lin, "Study on the Adversarial Sample Generation Algorithm Based on Adversarial Quantum Generation Adversarial Network," 2023 3rd International Symposium on Computer Technology and Information Science (ISCTIS), Chengdu, China, 2023, pp. 238-243, doi: 10.1109/ISCTIS58954.2023.10213103.
â‘¡Z. Liu, C. Li, F. Zheng, C. Huang and S. Zhang, "Enhancing Generalization Capability Through Local Features Reintegration for Adversarial Perturbations," 2024 6th International Conference on Communications, Information System and Computer Engineering (CISCE), Guangzhou, China, 2024, pp. 400-403, doi: 10.1109/CISCE62493.2024.10653439.
â‘¢ Fo Hu, Mengyuan Qian, Kailun He, Wen-an Zhang, Xusheng Yang. A Novel Multi-Feature Fusion Network with Spatial Partitioning Strategy and Cross-Attention for Armband-Based Gesture Recognition[J]. IEEE Transactions on Neural Systems and Rehabilitation Engineering. 2024, 32: 3878-3890.
â‘£Hu F, Zhang L, Yang X, et al. EEG-Based Driver Fatigue Detection Using Spatio-Temporal Fusion Network With Brain Region Partitioning Strategy[J]. IEEE Transactions on Intelligent Transportation Systems, 2024, 25(8): 9618-9630.
⑤M. M. Esfahani, M. H. Najafi and H. Sadati, "Optimizing EEG Signal Classification for Motor Imagery BCIs: FilterBank CSP with Riemannian Manifolds and Ensemble Learning Models," 2023 9th International Conference on Signal Processing and Intelligent Systems (ICSPIS), Bali, Indonesia, 2023, pp. 1-6, doi: 10.1109/ICSPIS59665.2023.10402664.
â‘¥M. Moein Esfahani and H. Sadati, "Application of NSGA-II in Channel Selection of Motor Imagery EEG Signals with Common Spatio-Spectral Patterns in BCI Systems," 2022 8th International Conference on Control, Instrumentation and Automation (ICCIA), Tehran, Iran, Islamic Republic of, 2022, pp. 1-6, doi: 10.1109/ICCIA54998.2022.9737199.
Comments on the Quality of English Language
-
Avoid repetition and redundancy: There are instances of repetition in the article, particularly in the introduction and discussion sections. It is recommended to use synonyms or rephrase sentences to avoid repetition and improve readability.
-
Vary sentence structure: The article often uses simple sentences, which can make the text monotonous. Consider incorporating more complex sentence structures, such as compound and complex sentences, to add variety and interest.
-
Pay attention to verb tense: Ensure consistent use of verb tenses throughout the article. For example, when discussing past research, use past tense, while for present research and future work, use present and future tenses, respectively.
Reviewer 2 Report
Comments and Suggestions for Authors
The study deals with the application of generative adversarial network (GAN) to EEG signals, in order to enhance the scalp EEG resolution using the information recorded from the foramen ovalis electrodes.
The model showed an accuracy of 76% to detect IEDs from scalp EEG respect to iEEG.
The paper is very interesting form the clinical point of view, the topic is timely (extract richer diagnostic insights from noninvasive sensor recordings) and the method employed is innovative, even if I am clinician and I am not so expert in models
In general, there the paper lack of clinical data of 18 patients analyzed; Are all they mesial temporal lobe epilepsy patients or mix epilepsy, have they structural etiology?
I am not sure about the patients selection but If all are MTLE I would highlight this point in the title or in methods in order to not generalized this results to all focal epilepsies.
Minor comments for the authors:
How many IEDs were visible both in scalp than iEEG? This from the clinical point of view is interesting to understand how much iEEG enhance the IEDs detection.
line 21: “The scEEG is a non-invasive recording method that lacks detailed temporal information” I do not understand the sentence; temporal resolution is one of the main strengths of EEG signal….
Line 24 “high temporal details since”. Here again, the sampling rate is almost the same for both techniques. Did they mean that iEEG give a very localized and specific information
Line 36 “Only 9 to 22%” I believe that this affirmation is true only for mesial temporal epilepsy, this should be clarified
Line 215 avoid “18 epileptics” and use persons with epilepsy
Line 226 please clarify how long is a sample “as IED segments before and after the point illustrated as the onset of IED”; IED may be of different duration so the sample are all of the same lengths of different lengths?
Round 2
Reviewer 1 Report
Comments and Suggestions for Authors
Thank you for your detailed responses to my previous comments and for the valuable additions to the manuscript. Overall, you have made significant improvements in explaining the model's principles, experimental results, and clinical applications. However, there are still a few key areas that need further clarification or enhancement to improve the quality of the paper. Below are my further suggestions and comments on your revised manuscript:
1. Although you have added a detailed description of the SPADE blocks, LSTM layers, and how latent space is used in the generator, there is still room for improvement, especially in the technical explanation of how SPADE retains spatial information and how latent space integrates with scEEG signals in the generation process. Please further clarify how SPADE differs from traditional normalization methods in terms of spatial information retention, particularly in the scEEG to iEEG mapping process. Regarding latent space, provide more details on how its compressed representation enhances the quality of generated iEEG and how latent space is integrated with scEEG signals.
2. While you mentioned that the inference time for VAE-cGAN is very short (<3 milliseconds), the computational complexity during training and comparisons with other models were not provided in detail. This is crucial for assessing the practical feasibility of the model. Please provide more details on the computational complexity during the training phase (e.g., GPU memory usage, training time) and compare it with existing models. Although the inference time is short, the computational cost during training remains a critical factor for real-world applications.
3. In your discussion of model limitations, you mention the dependency on consistent neuroanatomical data and individual differences. However, the issue of the model's sensitivity to noise, particularly noise sources commonly found in clinical environments (e.g., EMG interference, motion artifacts), has not been sufficiently discussed. Please add a more thorough analysis of the model’s sensitivity to noise, particularly common noise types in scEEG data (e.g., EMG interference, motion artifacts) and their impact on model performance. Propose possible solutions for addressing these challenges.
4. In your revised manuscript, you have included some of the recommended references, but several key recommended references were still not cited. These references are closely related to your research methodology and background, and their inclusion would strengthen the literature review section and the academic depth of the paper.
5. Upon reviewing the manuscript, I noticed that there is a relatively high level of repetition, particularly. This could affect the clarity and readability of the paper, and might also raise concerns regarding originality and rigor in the writing
Comments on the Quality of English LanguageN/A
